# Raman Spectroscopy of Disperse Systems with Varying Particle Sizes and Correction of Signal Losses

**DOI:** 10.3390/s24103132

**Published:** 2024-05-15

**Authors:** Erik Spoor, Viktoria Oerke, Matthias Rädle, Jens-Uwe Repke

**Affiliations:** 1CeMOS Research and Transfer Center, Mannheim University of Applied Sciences, Paul-Wittsack-Str. 10, 68163 Mannheim, Germanym.raedle@hs-mannheim.de (M.R.); 2Process Dynamics and Operations Group, Technische Universität Berlin, Straße des 17. Juni 135, 10623 Berlin, Germany; jens-uwe.repke@tu-berlin.de

**Keywords:** disperse phase, continuous phase, optical spectroscopy, particle measurement, process control, Raman spectroscopy, process engineering, UV/VIS spectroscopy, suspension measurement

## Abstract

In this paper, a dispersion of glass beads of different sizes in an ammonium nitrate solution is investigated with the aid of Raman spectroscopy. The signal losses caused by the dispersion are quantified by an additional scattered light measurement and used to correct the measured ammonium nitrate concentration. Each individual glass bead represents an interface at which the excitation laser is deflected from its direction causing distortion in the received Raman signal. It is shown that the scattering losses measured with the scattered light probe correlate with the loss of the Raman signal, which means that the data obtained can be used to correct the measured values. The resulting correction function considers different particle sizes in the range of 2–99 µm as well as ammonium nitrate concentrations of 0–20 wt% and delivers an *RMSEP* of 1.952 wt%. This correction provides easier process access to dispersions that were previously difficult or impossible to measure.

## 1. Introduction

Process analytical technology is a diverse field of measurement techniques for monitoring processes and analyzing the composition of mixtures [1,2,3]. Optical measurement technology, for example, offers direct access to process parameters such as the concentration of individual educts or products [4,5,6,7].

Frequently used optical measurement methods include UV/VIS, infrared, fluorescence and Raman spectroscopy. VIS spectroscopy is often used for the analysis of colors in order to achieve reproducible results [4,6,7]. In the UV and NIR spectrum, on the other hand, a structural analysis of samples is possible. If a substance has active bands in these ranges, changes or shifts can occur with concentration, allowing even the smallest concentrations to be detected [8]. Measurements in the NIR spectrum can require a great deal of calibration and are very sensitive to changing process parameters [9,10]. Another method is fluorescence spectroscopy, which is used in biomedicine, for example [11]. A sample is excited by light irradiation, which then emits a photon of a different wavelength. A major advantage of this measurement technique is its versatility and sensitivity, which means that several fluorophores within the same sample can be examined using different excitation wavelengths. However, this measurement method is limited to substances that emit photons [11,12].

Raman spectroscopy, which is used in this work, is a constantly developing field. The advantage of this technology is that non-contact and non-destructive measurements are possible through a glass pane, which makes process access easy to implement. In addition, Raman spectroscopy is molecule-sensitive and allows for the precise differentiation of different components, with very short measurement times in the range of seconds to milliseconds. In Raman spectroscopy, a target is irradiated by a laser, and the photons interact with the molecules. The valence electrons are thereby raised to an unstable energy level and then drop again, causing photons with altered frequencies, energies and wavelengths to be scattered in all spatial directions. The wavelength shift, also known as the Raman shift, is dependent on the molecular group and can therefore be used for the qualitative analysis of material compositions. In addition, the strength of the resulting signal is linearly dependent on the concentration, which allows for a quantitative evaluation of the target. The use of this measurement technique in homogeneous mixtures is now widespread and much researched [4,6,13,14,15,16]. However, the analysis of heterogeneous mixtures represents a special application. Disperse mixtures such as emulsions and suspensions have a high number of interfaces at which both the excitation light and the resulting Raman signal can be scattered. This reduces the power density of the laser at the focal point, leading to a strong attenuation of the measured signal and, therefore, to deviating measurement results or even results that cannot be interpreted [17,18,19,20,21]. A simple variant of signal correction is to compare the characteristic peaks of different components. However, this requires a constant component, such as a solvent, which can be used as a reference. In addition, the reference peak must lie within the measurement range of the spectrometer [21,22,23]. In the application example of this paper, a Tec5 (Steinbach, Germany) spectrometer is used that has a measuring range up to 3200 cm^−1^ but uses water as the solvent, which has its characteristic peak at 3400 cm^−1^. More complex calculation models are also used in the literature, which have many parameters to be calibrated but have the associated disadvantage of high computational effort and time-consuming calibration [24,25]. There are also other approaches to solving the problems of measuring dispersed phases, such as refractive index matching. Here, an additional component is added to a phase, which reduces or even completely removes the refractive index difference between the phases. As a result, optical diffraction no longer takes place. However, adding a further component to the system can lead to new problems, such as the overlapping of peaks or the later purification of the product, and is therefore not universally applicable [18,26].

In order to have a simple and more universally applicable correction method, an additional probe is used in this paper that quantifies the losses of the excitation beam. This makes it possible to establish a direct correlation between reduced Raman peaks and increasing light scattering with increasing particle concentration and, finally, to correct the Raman spectra. For this purpose, the concentration of an ammonium nitrate solution in which glass beads are added as a disperse phase is investigated. The glass beads do not have a spectrum of their own that collides with the signal of the ammonium nitrate, which means that the focus of the observation can be placed on the investigation of the scattering effects caused by dispersed particles.

## 2. Materials and Methods

The basic principle of the measurements is that a Raman probe excites the sample and detects the Stokes signal and, at the same time, a scattering probe detects the losses of the excitation laser at an increased particle concentration. In a fluid without scattering losses due to particles or droplets, the laser passes through the fluid in a straight line, and no excitation photons should reach the scattering probe. Also, no significant amount of the Stokes signal is detected from the Raman measurements. This is partly because the scattering probe is not focused and partly because the detection module used is not designed for Raman measurements and has a much lower sensitivity. As soon as particles are present in the sample, the excitation light from the laser is scattered and detected by the probe. This effect and the correlation with the Raman signal are investigated and used for signal correction.

The measurement setup consists of a macro quartz glass cuvette from Hellma (Müllheim, Germany) with a sample that is placed in the focal point of two probes and positioned on a magnetic stirrer (Figure 1). An InPhotonics (Norwood, MA, USA) RPS785/16-5 probe was used to excite the sample. It has a focus length of 7.5 mm with a spot size of 158 µm and a numerical aperture of 0.27. The probe is connected to a Raman spectrometer MultiSpec Desk ETH with an LS-LD laser cassette (785 nm excitation wavelength) and an SC-CCD RAMAN spectrometer cassette (detection range 319–3213 cm^−1^ with increments of 1 cm^−1^) from tec5 (Steinbach, Germany). The laser power can be set between 50 and 500 mW and was adjusted to the maximum output for the measurements. The detector is a scientific thermoelectric-cooled CCD detector with a spectral resolution of 7 cm^−1^. The position of the probe was adjusted so that the focal point was approx. 2 mm inside the cuvette. Normally, measurements would be taken in the center of a cuvette, but previous studies [27] have shown that a lower penetration depth is beneficial for the signal at higher particle concentrations. The second probe is a self-built scattering probe consisting of seven 200 µm fibers arranged in a circle. It has already been tested in other applications and has proven to be robust and cost-effective in the use of scattered light measurements [28]. This probe does not emit its own light, but it is only used to detect scattered light. For this purpose, it is connected to an MCS 601 UV-NIR C spectrometer from Zeiss (Oberkochen, Germany) (detection range of 190–1015 nm with increments of 0.5 nm) with a spectral resolution of 2.4 nm. The inner components consist of a grating with a diode array.

In order to ensure the consistency of the measurements, a mount was manufactured that allows the Raman probe to be placed vertically to the cuvette at the desired distance and the scattered light probe to be attached aligned with the focal point of the Raman probe (Figure 2).

The basic substances under investigation have already been examined in a previous paper [27]. This involves an ammonium nitrate solution into which glass beads (SiO_2_) are dispersed. To improve the understanding of disperse systems and their influence on Raman spectroscopy, four different particle sizes were investigated. The particles used were Omicron NP3 (2.093 µm), Omicron NP5 (4.089 µm), Micropearl (6.604 µm) and Starmixx (99.149 µm). The particle sizes refer to the Sauter diameter, which was measured with a HELOS particle size analyzer from Sympatec (Clausthal-Zellerfeld, Germany) [29]. Five ammonium nitrate solutions with deionized water were prepared for each of the four particle sizes. For each ammonium nitrate solution, 12 glass bead concentrations were then prepared. This corresponds to a total number of 240 samples. The individual concentrations are listed in Table 1 for one particle size at one ammonium nitrate concentration. It is expected that the size of the particles will have a measurable influence on the Raman signal. For example, at the same concentrations, there is a larger number of NP3 particles in the same volume than would be the case with Starmixx. As the number of particles increases, so does the number of interfaces at which light refraction can occur. This means that with increasing particle size, less refraction and therefore signal loss in the Raman spectrum is to be expected. This influence is analyzed in more detail in the following measurements using the selected particle sizes and concentrations shown in Table 1.

## 3. Results

Initially, the peak of the Raman spectrum to be examined must be identified. As an example, the spectra of 20 wt% ammonium nitrate with no particles and with 3 wt% particles are shown in Figure 3. Ammonium nitrate (NH_4_NO_3_) consists of NH_4_^+^ and NO_3_^−^ ions, which can be identified in the Raman spectrum by their characteristic bands. In the spectrum shown, the NO_3_^−^ ions generate relatively weak signals at 716 cm^−1^, 1390 cm^−1^ and 1671 cm^−1^ and a strong signal at 1047 cm^−1^, which is most suitable for identifying the ammonium nitrate concentration. NH_4_^+^ only produces a weak signal at 1460 cm^−1^, which overlaps with the signal of the NO_3_^−^ ions [30,31,32]. The NO_3_^−^ peak at 1047 cm^−1^ is therefore best suited for evaluation. This peak has a FWHM (full width half maximum) of 11 cm^−1^ and is detectable both with and without 3 wt% particles. All intensities of the Raman signal that were measured refer to this peak. The measurement signal of the water content cannot be evaluated in these spectra, as the significant OH peak is at approx. 3400 cm^−1^ and only a small part of the side at 3200 cm^−1^ is recognizable [33,34]. The glass beads themselves do not generate a signal of their own that can be measured in the spectrum.

All measured data are processed immediately after acquisition in order to minimize offset errors between the measurement series. In the first step, a baseline correction is performed in Formula (1) by subtracting the mean values of the left edge (1107 cm^−1^) and right edge (1987 cm^−1^) from the intensity value of the ammonium nitrate peak.
(1)$$I_{AN}=I_{{1047cm}^{-1}}-\frac{I_{{1107cm}^{-1}}+I_{{1987cm}^{-1}}}{2}$$

After the baseline correction, the database is normalized, as there may be a slight displacement of the probes between the measurements. The measurements at 0 wt% particles can serve as a reference. These measuring points should provide the same values for the same ammonium nitrate concentration regardless of the defined particle type. For correction, all data for each ammonium nitrate concentration are related to the measured values of Micropearl (6.604 µm) at 0 wt% particles (Formula (2)).
(2)$$I_{ANnorm}=\frac{I_{AN}\cdot I_{AN}(6.604\;\mu m;\;0\;wt\%\;particles)}{I_{AN}(0\;wt\%\;particles)}$$

All Raman data of the ammonium nitrate peak are plotted in Figure 4 sorted by particle size over the particle concentration. Because of the previous normalization, the 0 wt% measurement points of all particle sizes and the same ammonium nitrate concentration correspond to each other. In principle, there is a linear correlation between a Raman signal and the concentration of the target. Examining the measurement points at a constant particle concentration, it can be seen that this correlation exists for the ammonium nitrate concentration. However, the observation over the particle concentration provides a steep course up to 0.25 wt% and then a flattening intensity curve. The measurement series of NP3 and NP5 run into saturation towards 3 wt% and then no longer show any significant changes with the particle concentration. The curves of Micropearl and Starmixx, on the other hand, do not yet show any saturation. As the particle size increases, the signal losses of the measurement decrease. While the peak heights for NP3 decrease by a factor of 6 to 10, the peaks for Starmixx only decrease by a factor of 1.6 to 2. The reduced light scattering with increasing particle size can be explained by the fact that, overall, larger particles have a smaller number of boundaries than small particles at the same concentration.

The measurements with the scattering probe are shown as an example in Figure 5, using 20 wt% ammonium nitrate with no particles and with 3 wt% particles.

The produced peak is a combination of Rayleigh scattering, which is scattered in all spatial directions during molecular interactions, and direct excitation radiation, which is deflected by the particles. A differentiation is not possible since both signals have the same wavelength. The peak is located at 785 nm with a FWHM of 2.5 nm, and the spectrum shows no other peaks. With 0 wt% particles, the peak height is approx. 9000 counts and is therefore 333 times smaller than the peak with 3 wt% particles. This shows, on the one hand, that light scattering occurs on the cuvette and in the medium even without particles and, on the other hand, that the light scattering mentioned is not significant compared with the scattering caused by the particles.

The data are calculated in the same way as in Formula (2) for the Raman data. A baseline correction is not necessary as the baseline has no relevant influence compared with the peak height. The peak height is calculated using Formula (3), which is based on the Beer–Lambert [4] law but uses the reciprocal of the intensity ratio. Typically, the Beer–Lambert law would be used for transmission measurements to calculate the light loss as extinction. For a transmission measurement, the intensity would decrease with the particle concentration, which corresponds to an increase in extinction. However, as the measurements of the scattering probe initially measure the opposite effect, that is, the increase in the measurement signal, as more light is scattered by the particles, the reciprocal value must be used for the purpose of representation in order to continue to depict an increase in light loss or extinction with increasing particle concentration. This is performed in order to have a better representation of the effects occurring and, in addition, to be comparable to the previous paper where an actual transmission setup was used [27].
(3)$$E_{\lambda }=\log_{10}\left(\frac{I}{I_{0\;wt\%\;particles}}\right)$$

Analogous to the Raman data, the scattered light data are sorted by particle size and plotted over the particle concentration (Figure 6). The calculation according to the Beer–Lambert law, which takes the measurement points with 0 wt% particles as a reference, results in an origin in the zero points of the graphs for all measurement series. The curves are comparable to those of the Raman measurements, as here, too, the measurement signal changes very strongly up to 0.25 wt% and then becomes saturated. The correlation can also be logically explained by the fact that the more laser radiation is scattered, the lower the excitation of the sample and the intensity of the Stokes scattering. The correlation can also be seen in the fact that the light scattering decreases with increasing particle size.

One difference to the Raman data is that no dependence of the scattered light on the ammonium nitrate concentration can be measured. This is also to be expected, as the laser radiation should not contain any information about it. The variance in the measured values at a constant particle concentration is on average approx. 0.09 for NP3 and approx. 0.16 for Starmixx. The gradient of the measured value fluctuations with the particle size could be due to the fact that larger particles sink faster and thus have a greater dependence on the stirrer speed.

Although the representation of the signals via the particle concentration allows for an explanation of the scattering effects, it provides an incomplete picture of the relationships. This is due to the fact that the concentration does not contain any information about the size of the particles. In order to obtain a more accurate picture of the relationships, the specific disperse surface of all samples is calculated. This value is defined as the total surface area of all particles per volume. First, the total density of the mixture $\rho_{t}$ must be calculated from the mass fraction (w) and individual densities (ρ) of ammonium nitrate (AN), distilled water (dw) and particles (p) according to Formula (4):(4)$$\rho_{t}=\frac{1}{\frac{w_{AN}}{\rho_{AN}}+\frac{w_{dw}}{\rho_{dw}}+\frac{w_{p}}{\rho_{p}}}$$

The total number of particles ($N_{p}$) can be determined in Formula (5) from the weighed mass ($m_{p}$), the density ($\rho_{p}$) and the diameter (D) of the particles:(5)$$N_{p}=\frac{m_{p}}{\rho_{p}\cdot \frac{\pi \cdot D^{3}}{6}}$$

The specific disperse surface ($S_{d}$) can then be determined from the number of particles ($N_{p}$), mixing density ($\rho_{t}$), total weight of the solution ($m_{t}$) and surface area of a single particle (S) (Formula (6)):(6)$$S_{d}=S\cdot N_{p}\cdot \frac{\rho_{t}}{m_{t}}$$

The measured values of all particle sizes at 20 wt% ammonium nitrate are plotted against the specific disperse surface in Figure 7. The new plot shows that all measurement series follow the same course but are distorted when plotted against the concentration. The linearity of the Starmixx measurements can also be explained, as all data fall off in an approximately linear pattern up to a specific disperse surface of approx. 3000 m^2^/m^3^. The previously described effect that larger particles at the same concentration have fewer interfaces than smaller particles is now reflected in the new plot, whereby at 3 wt%, Starmixx has a specific disperse surface of 810 m^2^/m^3^ and NP3 of 37,930 m^2^/m^3^.

To view all the measurement data in a collective context, the Raman data are plotted as a change in peak height (compared to the 0 wt% particle measurement) against the corresponding scattered light value in Figure 8. A trend comparable to the plot against the specific disperse surface can be seen, where the Starmixx data are distributed over lower *x*-axis values than the remaining data sets. For the NP3, NP5 and Micropearl data points, the spread across the *x*-axis also correlates with the particle size. The clear difference to Starmixx is due to the particle size being up to 47 times larger. Nevertheless, the overall picture shows a clear trend between the peak height change and light scattering. Especially in the range below a scattering of 1, the deviations of 0.02 from the trend are quite small. Above 1, the curve becomes more of a scatter plot and shows a deviation of 0.57 from the trend. The diagram demonstrates the direct correlation between Raman peak and scattering losses, making the measurement data independent of any knowledge of the particle size or specific dispersed surface area. All factors influencing the measurement signal are reflected in the form of the scattering loss.

To obtain a regression that can be used to correct the Raman values, the amount of data is first reduced by taking the mean values from the same particle concentration (Figure 9a). Starting from this, different trend lines are adjusted iteratively until the highest possible regression is achieved. In order to obtain a good adaptation, it is necessary to split the data at a scatter value of 0.9 and to describe the data below 0.9 by a second-degree polynomial function and the data above 0.9 by a third-degree polynomial function. The trend lines and formulas obtained are shown in Figure 9b.

From the formulas shown in Figure 9b, the change in peak height can be calculated as the correction factor ∆I. If this is multiplied by the measured intensity $I_{AN}$ in Formula (7), the result is the corrected intensity $I_{AN-corr}$, which should match a measured value without particles:(7)$$I_{AN-corr}=I_{AN}\cdot ∆I$$

For a clear display, the values are converted into mass fractions of the predicted ammonium nitrate concentration ($\omega_{AN}$), which can be performed using the proximity Equation (8). The relationship originates from the regression of the measured Raman data without particles:(8)$$\omega_{AN}=9\cdot 10^{-7}\cdot I_{AN-corr}+0.0023$$

The mean corrected ammonium nitrate concentrations of the measurement series are shown in Figure 10. A theoretically optimal correction should lie on the corresponding horizontal grid line at 0, 5, 10, 15 and 20 wt%, respectively. All deviations from this are errors in the prediction. As a measure of the error, the *RMSEP* (Root Mean Squared Error of Prediction) is calculated according to Formula (9) [35], plotted as an error indicator in Figure 10 and listed individually in Table 2.
(9)$$RMSEP=\sqrt{\frac{1}{N}\cdot \sum_{i=1}^{N}{\left(y_{i}-{\bar{y}}_{1}\right)}^{2}}$$

The highest deviation with an *RMSEP* of 5.386 wt% occurs with NP5 and 15 wt% ammonium nitrate. The region at 0.1–0.2 wt% particles is significant for this, as the deviation there ranges from 7.2 to 10.2 wt%. The Raman data and scattered light data initially show no clear anomalies. Figure 8 shows that the NP5 data tend to be lower than other data points and are therefore below the regression curve. This results in an over-correction, which can be seen in the data points in question. The reverse is true for Micropearl at 10 wt% ammonium nitrate, for example, where the data points are above the regression curve and under-correct the prediction by 3.2–4.1 wt%. The best predictions occur in the range of the 0 and 5 wt% ammonium nitrate concentrations, where the deviation is as low as 0.25 wt% for Starmixx at 0 wt% ammonium nitrate and 0.311 wt% at 5 wt% ammonium nitrate. Overall, this results in an *RMSEP* of 1.176–2.896 wt% for the average observation per particle size and an *RMSEP* of 1.952 wt% when considering all measurement data. As mentioned before, the lower concentrations have a much lower *RMSEP* compared with the average value and some outliers have a higher value. But considering that the signal approaches zero without the correction, an overall averaged *RMSEP* of 1.952 wt% of the prediction gives a good indication of the actual value of the data. While this accuracy is quite high for precise concentration determinations and compared with homogeneous mixtures, the accuracy is well suited for reaction tracking or trend measurements.

## 4. Discussion

The scope of this work was to investigate the relationship between Raman measurements and dispersive systems based on a simple system of ammonium nitrate solution and glass beads and to find a way to correct the signal losses. In the previous work [27], this was performed with only one constant particle size by taking a separate transmission measurement with a separate light source. This measurement was a reference for how much light is lost due to dispersion. This has now been optimized by first replacing the transmission measurement with a scattered light probe, which measures the laser radiation, and then performing both measurements simultaneously. In addition, the substance system was expanded to include more particle sizes to obtain a more complex assessment.

The measurements of 2–99 µm particles and 0–3 wt% particle concentrations in 0–20 wt% ammonium nitrate solutions show the relationship between increasing particle concentration, decreasing Raman signal and increasing scattered light signal. The Raman data of NP3 (2.093 µm), NP5 (4.089 µm) and Micropearl (6.604 µm) provide a steep, almost linear decrease up to 0.25 wt% and then change into a flattening curve that runs almost parallel to the *x*-axis at 3 wt%. The Starmixx (99.149 µm) data, on the other hand, show a deviating curve over the particle concentration, as the data points fall almost linearly and show significantly lower losses. This can be explained by plotting the measured data over the specific disperse surface area.

It can be concluded that the 99.149 µm particles at the same concentration as the 2.093 µm particles have a significantly lower specific disperse surface area, which means that there are fewer boundary surfaces and, therefore, less signal interference can occur. This plot also shows that the same course of signal loss occurs for all particle sizes. For this reason, a representation via the concentration only provides a distorted image and delivers an incomplete picture. The Raman data were then plotted over the scattered light data as a change in peak height to obtain a correction function. This plot also illustrates the relationship between scattered light data and specific dispersive surface area, as the plots show the same distribution of data across the *x*-axis. Starmixx provides significantly lower scattered light values because of its lower specific dispersive surface area (approx. 1.5 at approx. 800 m^2^/m^3^) and NP3 provides the highest scattered light values as it has the largest specific dispersive surface area (approx. 2.7 at approx. 38,000 m^2^/m^3^). This correlation between scattered light and specific dispersive surface area shows that the scattered light measurement is a good measure and indirectly contains information on particle size and concentration. 

The resulting correction was then applied to all measurement data, resulting in an average *RMSEP* for all measurement data of 1.952 wt%. However, deviations of up to 10 wt% can occur when considering individual data points. This is due to the fact that a generally applicable model was used, which can be applied to all measurement data, but is therefore also more susceptible to deviating measurement series. An average *RMSEP* of 1.952 wt% appears to be quite high for low concentrations, since in the example of 5 wt% measurements, this would represent a deviation of 40% of the measured value. On closer inspection, however, the deviations in this range are significantly lower, with an *RMSEP* of 0.311–1.072 wt% depending on the particle size under consideration. Even though for some data points, such as 15 wt% ammonium nitrate with NP5 particles, the prediction shows significant deviations, it still reflects values in the correct order of magnitude. The mean *RMSEP* is therefore not representative of every prediction of the ammonium nitrate concentration but provides a rough overview for a broad application of the model. Compared with the measurement of homogeneous mixtures, an average *RMSEP* of 1.952 wt% can be quite high, but the accuracy is still sufficient to carry out reaction tracing or investigations of concentration trends. A more specific regression and a lower *RMSEP* for individual ammonium nitrate concentrations or particle sizes would be conceivable but would have the disadvantage of greater calibration effort and would require knowledge of the ammonium nitrate concentration or particle size. However, despite deviations in individual measurement data, it can be shown that a correction is possible using this measurement technique. In further work, the findings of this study will be transferred from suspensions to emulsions and applied to more complex systems.

## 5. Conclusions

It can be concluded that the performed correction is a good and simple option to correct measurement data of the continuous phase that are influenced by disperse systems. The scattered light database contains the required information on particle size and concentration and achieves an *RMSEP* of 1.952 wt% when considering all measurement data. This means that no further specific knowledge of the size or number of particles is required.

## Figures and Tables

**Figure 1 sensors-24-03132-f001:**
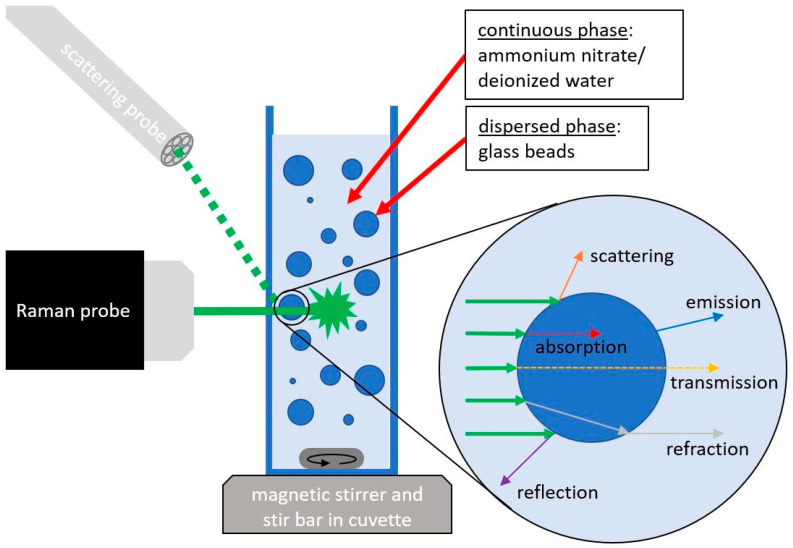
Scheme of the measurement setup with Raman and scattering probes, a cuvette with suspension on a magnetic stirrer and interaction of light.

**Figure 2 sensors-24-03132-f002:**
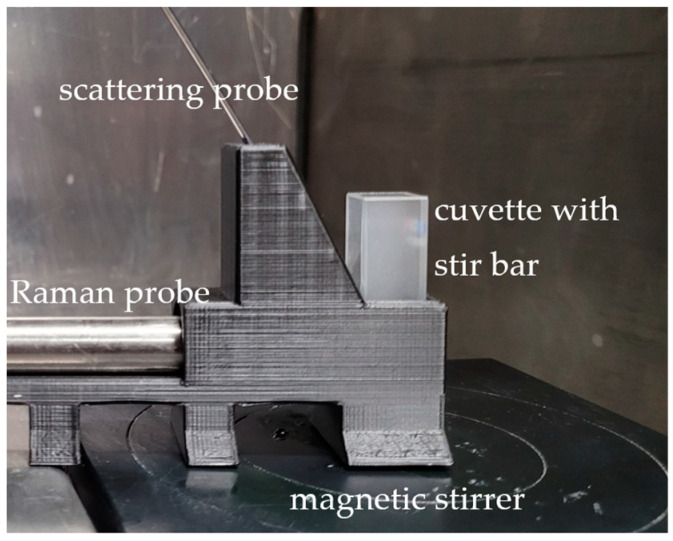
Setup of the Raman and scattering probes and the cuvette on the magnetic stirrer.

**Figure 3 sensors-24-03132-f003:**
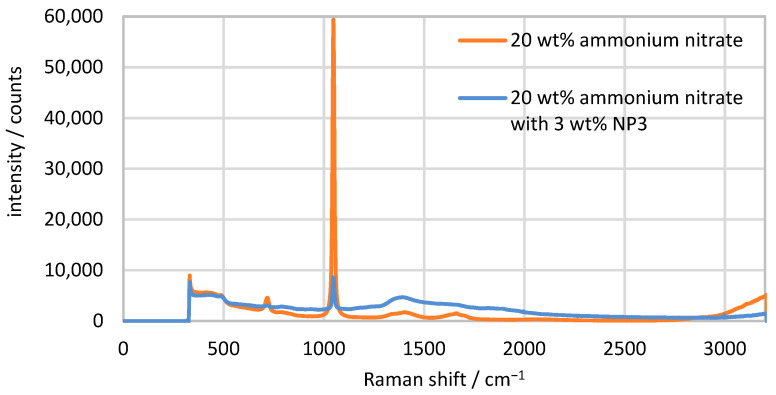
Spectra of the ammonium nitrate solution with 3 wt% NP3 particles and without particles for comparison.

**Figure 4 sensors-24-03132-f004:**
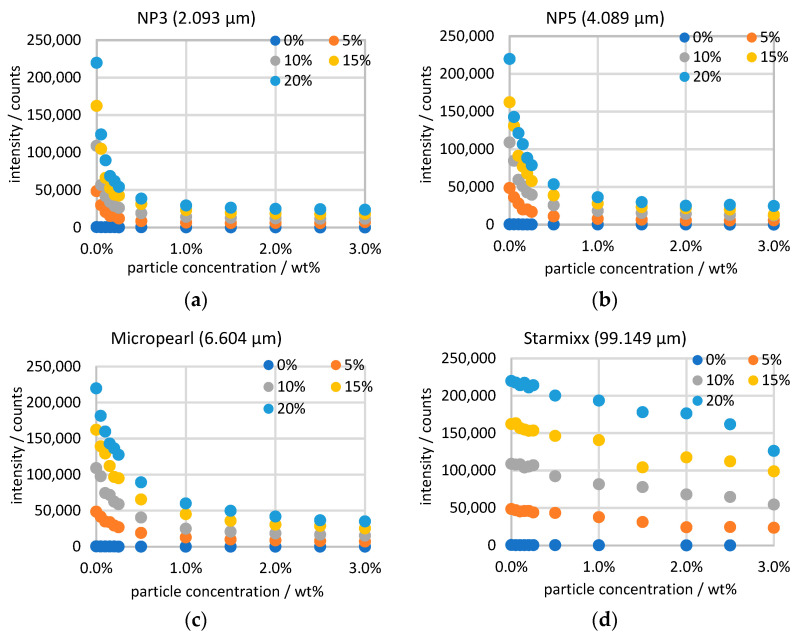
Raman measurements of particle sizes of (**a**) 2.093 µm; (**b**) 4.089 µm; (**c**) 6.604 µm and (**d**) 99.149 µm with increasing particle and ammonium nitrate concentrations.

**Figure 5 sensors-24-03132-f005:**
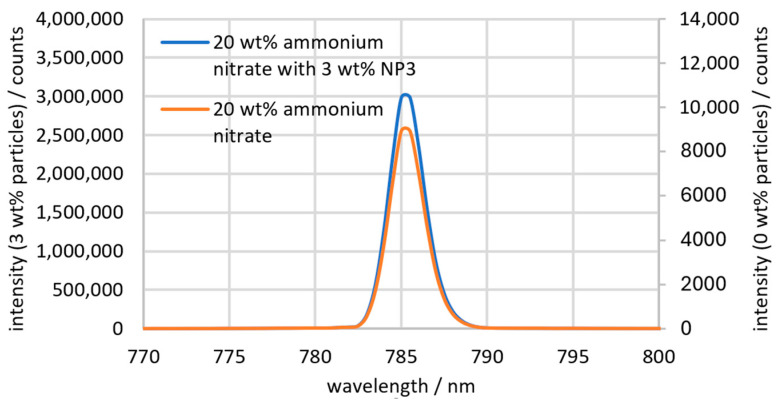
Peak of the excitation laser measured by the scattering probe with 3 wt% NP3 particles and without particles for comparison.

**Figure 6 sensors-24-03132-f006:**
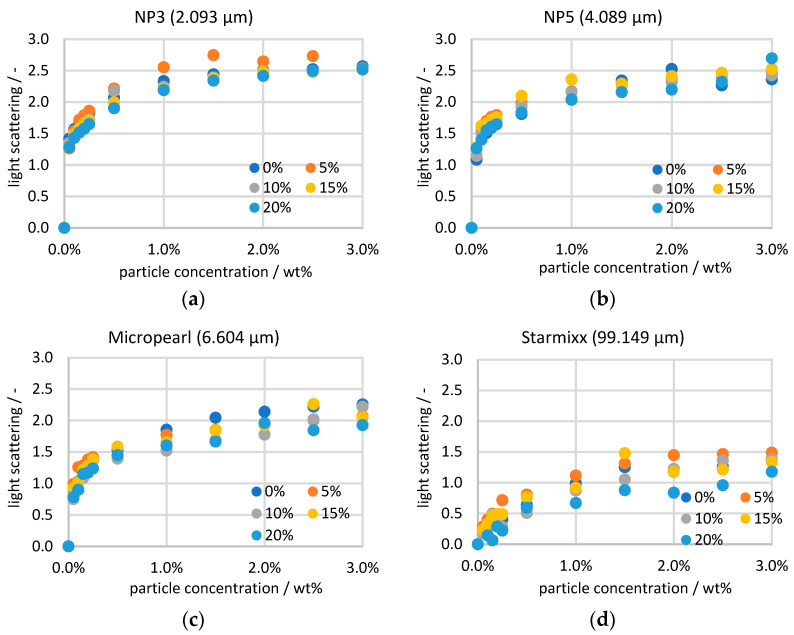
Scattering measurements of particle sizes of (**a**) 2.093 µm; (**b**) 4.089 µm; (**c**) 6.604 µm and (**d**) 99.149 µm with increasing particle and ammonium nitrate concentrations.

**Figure 7 sensors-24-03132-f007:**
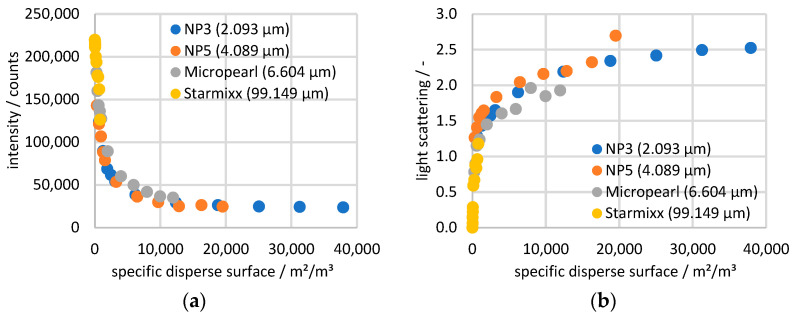
(**a**) Raman and (**b**) scattering data of all four particle sizes with the 20 wt% ammonium nitrate solution plotted over the specific disperse surface of the particles.

**Figure 8 sensors-24-03132-f008:**
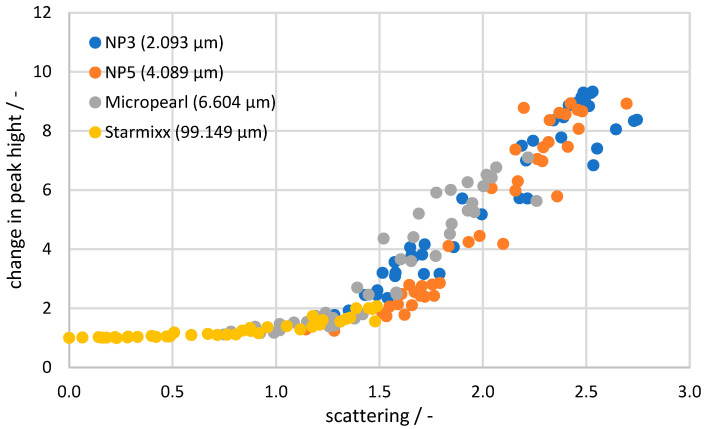
Change in the peak height of all samples plotted over the scattering of light, sorted by particle size.

**Figure 9 sensors-24-03132-f009:**
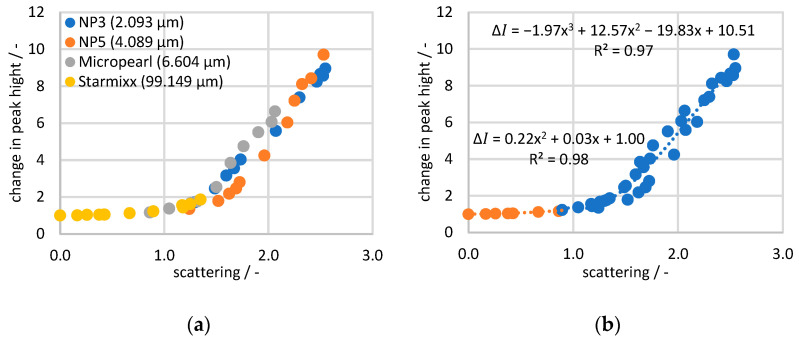
Average change in the peak height per particle concentration plotted over the average scattering of light per particle concentration: (**a**) sorted by particle size and (**b**) sorted by best regression with regression formulas.

**Figure 10 sensors-24-03132-f010:**
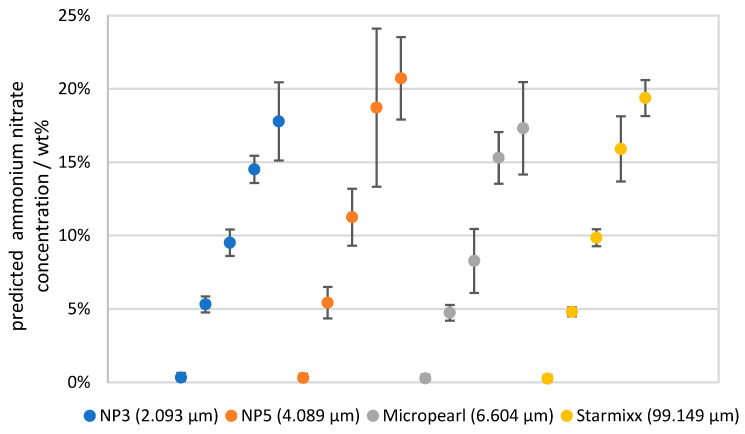
Calculated prediction for the ammonium nitrate concentration based on the measured scattering of light with increasing particle concentration and plotted *RMSEP* as an indicator of the prediction error.

**Table 1 sensors-24-03132-t001:** Overview of the concentrations of ammonium nitrate solutions and particles.

Particle Name (Size)	Ammonium Nitrate Solution/wt%	Particle Concentrations per Ammonium Nitrate Concentration/wt%
NP3 (2.093 µm)	0–20 in 5 wt% steps	0.00–0.25 in 0.05 wt% steps
		0.50–3.00 in 0.50 wt% steps
NP5 (4.089 µm)	0–20 in 5 wt% steps	0.00–0.25 in 0.05 wt% steps
		0.50–3.00 in 0.50 wt% steps
Micropearl (6.604 µm)	0–20 in 5 wt% steps	0.00–0.25 in 0.05 wt% steps
		0.50–3.00 in 0.50 wt% steps
Starmixx (99.149 µm)	0–20 in 5 wt% steps	0.00–0.25 in 0.05 wt% steps
		0.50–3.00 in 0.50 wt% steps

**Table 2 sensors-24-03132-t002:** Calculated *RMSEP* for all predicted ammonium nitrate concentrations.

Particle Name (Size)	AN-Conc./wt%	*RMSEP* per AN-Conc./wt%	*RMSEP* per Particle Size/wt%	Overall Averaged *RMSEP*/wt%
NP3 (2093 µm)	0	0.325	1.356	1.952
	5	0.544		
	10	0.906		
	15	0.925		
	20	2.668		
NP5 (4089 µm)	0	0.296	2.896	
	5	1.072		
	10	1.942		
	15	5.386		
	20	2.815		
Micropearl (6604 µm)	0	0.260	1.906	
	5	0.538		
	10	2.183		
	15	1.761		
	20	3.154		
Starmixx (99,149 µm)	0	0.250	1.176	
	5	0.311		
	10	0.578		
	15	2.219		
	20	1.225		

## Data Availability

The data presented in this study are available on request from the corresponding author.

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
