# Peer review of "Raman Spectroscopy of Disperse Systems with Varying Particle Sizes and Correction of Signal Losses"

_sensors, 2024, doi:10.3390/s24103132_

Round 1
Reviewer 1 Report
Comments and Suggestions for Authors
The manuscript must be improved before publication.

Only minor modifications.
Reviewer 2 Report
Comments and Suggestions for Authors
A correction method for Raman spectral signal loss in a variable particle size dispersion system is proposed in this manuscript. The proposed method can effectively solve the interference caused by sample particle size on Raman spectra, and it is believed that this manuscript will receive attention from Raman related research and applications. It is recommended that the manuscript be accepted after clarification of the comments.
1.The introduction should complement other relevant monitoring techniques for process analysis and highlight the advantages of Raman spectroscopy in this field. In addition, correction methods related to Raman spectroscopy should also be mentioned and completed in this section.
2.The second part should further analyse the reasons for the influence of particle size on Raman spectroscopy.
3. In the study of this method, the author selected four sample sizes. How does this method perform in correcting Raman spectroscopy for other sample sizes?
4. If the best regression equation is shown in Figure 9, is the fitted line consistent with the characteristic peaks of the sample?
5. Is the rightmost column in Table 2 the average RMSEP? This should be shown in the Table 2.
Comments on the Quality of English LanguageMinor editing of English language required.
Reviewer 3 Report
Comments and Suggestions for Authors
See attached file.

Round 2
Reviewer 1 Report
Comments and Suggestions for Authors
Modifications are adequate. The manuscript should be published